# Insulin signaling regulates R2 retrotransposon expression to orchestrate transgenerational rDNA copy number maintenance

Jonathan O. Nelson [1,2,3] ✉, Alyssa Slicko [2,3], Amelie A. Raz [2,3] &
Yukiko M. Yamashita [2,3,4] ✉

Preserving a large number of essential yet highly unstable ribosomal DNA (rDNA) repeats is critical for the germline to perpetuate the genome through generations. Spontaneous rDNA loss must be countered by rDNA copy number (CN) expansion. Germline rDNA CN expansion is best understood in *Drosophila melanogaster*, which relies on unequal sister chromatid exchange (USCE) initiated by DNA breaks at rDNA. The rDNA-specific retrotransposon R2 responsible for USCE-inducing DNA breaks is typically expressed only when rDNA CN is low to minimize the danger of DNA breaks; however, the underlying mechanism of R2 regulation remains unclear. Here we identify the insulin receptor (InR) as a major repressor of R2 expression, limiting unnecessary R2 activity. Through single-cell RNA sequencing, we find that male germline stem cells (GSCs), the major cell type that undergoes rDNA CN expansion, have reduced *InR* expression when rDNA CN is low. Reduced InR activity in turn leads to R2 expression and CN expansion. We further find that dietary manipulation alters R2 expression and rDNA CN expansion activity. This work reveals that the insulin pathway integrates rDNA CN surveying with environmental sensing, revealing a potential mechanism by which diet exerts heritable changes to genomic content.

Ribosomal DNA (rDNA) loci are essential regions of the genome containing hundreds of tandemly repeated ribosomal RNA (rRNA) genes. The repetitive nature of rDNA loci makes them prone to undergo intrachromatid recombination between rDNA copies, leading to rDNA copy number (CN) reduction[1] (Fig. 1A). Cell viability relies on a high number of rDNA repeats, thus protection against continual rDNA CN reduction is critical, especially in the lineages that continue through long time scale, such as the metazoan germline that passes the genome through generations[2]. In these cell types, the expansion of rDNA CN plays a critical role in

counteracting spontaneous rDNA CN reduction and achieving long-term rDNA CN maintenance[3,4].

*Drosophila melanogaster* and budding yeast have long served as excellent models to investigate rDNA CN expansion, with germline rDNA CN expansion first being described in *Drosophila* in the 1960s as the phenomenon called 'rDNA magnification'[5,6]. *Drosophila* that harbors unusually low rDNA CN exhibits visible phenotypes such as thin bristles and cuticle defects, collectively called the 'bobbed' (bb) phenotype[7]. rDNA magnification describes the process of bobbed fathers to produce offspring that have reverted to wild-type cuticles

[1]Department of Biochemistry and Cell Biology, Stony Brook University, Stony Brook, NY, USA. [2]Whitehead Institute for Biomedical Research, Cambridge, MA, USA. [3]Howard Hughes Medical Institute, Cambridge, MA, USA. [4]Department of Biology, MIT, Cambridge, MA, USA.
✉e-mail: jonathan.nelson@stonybrook.edu; yukikomy@wi.mit.edu

**Fig. 1 | Single-cell RNA sequencing reveals candidate repressors of rDNA magnification. A** Model of R2 function in rDNA copy number (CN) maintenance. Expression of R2 when rDNA CN is reduced causes rDNA-specific R2 endonuclease activity to create double-stranded DNA breaks (DSBs) at the rDNA locus, which can lead to rDNA CN expansion by unequal sister chromatid exchange (USCE) during their repair. **B** UMAP 2-dimensional reduction of early germ cells and somatic cyst cells from combined low ($bb^{z9}/Ybb^0$; *UAS-upd*/+; *nos-Gal4*/+) and normal rDNA CN *upd* over-expression testes ($bb^{z9}/Ybb^+$; *UAS-upd*/+; *nos-Gal4*/+). Cell type clusters are germline stem cells (GSCs), spermatogonia (SG), and cyst stem cells (CySC). **C** Differential gene expression in GSCs from combined analyses. The lowest fold change and highest *p*-value produced for either analysis are displayed. Significance for differential gene expression analysis was determined by a non-parametric Wilcoxon rank sum test and adjusted for multiple comparisons using Bonferroni correction. **D** Differential gene expression in low rDNA CN GSCs and SG determined by cluster-based cell selection. Significant gene expression change indicated by a log$_2$ fold change >0.25 or <−0.25. Source data are provided as a Source Data file.

due to rDNA CN expansion within the father's germline[5]. Recent studies suggest that rDNA magnification is a manifestation of the rDNA CN expansion mechanism that maintains rDNA against spontaneous CN reduction in the germline[3,4]. The easily scorable cuticle phenotype of bobbed flies and its reversion due to rDNA CN expansion has served as a powerful paradigm for studying germline rDNA CN expansion[8].

rDNA magnification is thought to occur by unequal sister chromatid exchange (USCE) (Fig. 1A)[8,9]. USCE is triggered by DNA double-strand breaks (DSBs), which leads to homology-dependent recombinational repair between misaligned copies, allowing one sister chromatid to acquire rDNA copies from the other sister chromatid (Fig. 1A)[9]. We recently found that an rDNA-specific retrotransposon R2 is responsible for DSB formation during magnification (Fig. 1A)[4]. R2 was shown to be required to maintain rDNA copy number through generations[4], representing a striking example of host-transposable element (TE) mutualism, wherein active TEs benefit the host and are required for species survival[4,10]. Whereas R2's ability to create DSBs at rDNA loci is required for rDNA magnification[4], unnecessary R2 transposition and DSB formation at rDNA threaten the stability of rDNA loci[11]. Therefore, for R2 to be beneficial to the host, there must be a mechanism that regulates R2 activity according to the host's needs, such as when rDNA CN is critically low or physiological demand for rDNA copies is increased.

To achieve effective rDNA magnification, USCE occurs in asymmetrically dividing germline stem cells (GSCs), such that the sister chromatid that gained rDNA CN can be selectively retained within the GSC[12,13]. This biased inheritance in USCE products during asymmetric GSC divisions can effectively expand rDNA CN through repeated rounds of USCE over successive GSC divisions[13]. Additionally, R2

expression during rDNA magnification is restricted to GSCs, whereas differentiating germ cells (spermatogonia (SGs)) do not express R2 to avoid DSBs in these cells that are particularly sensitive to DNA damage[13,14]. Therefore, GSCs appear to be uniquely capable of regulating R2 expression in response to rDNA CN to control the activity of rDNA magnification. It remains unknown how these cells sense the need for rDNA CN expansion and how such information leads to R2 derepression.

Here, using single-cell RNA sequencing (scRNA-seq), we identified genes that are differentially regulated in GSCs in response to low rDNA CN. Through this analysis, we identified Insulin-like Receptor (InR) as a gene downregulated in GSCs under low rDNA CN. We found that InR is a negative regulator of R2 expression, thereby preventing unnecessary R2 expression when cells have sufficient rDNA CN. Further, we showed that the mechanistic target of rapamycin (mTor), a major effector of insulin/IGF signaling (IIS), also represses R2 expression. We propose that IIS transduces rDNA CN surveillance within GSCs to the regulation of rDNA magnification activity via its ability to repress R2 expression.

## Results

### Single-cell RNA sequencing identifies differentially expressed GSC genes upon rDNA CN reduction

We recently showed that rDNA magnification primarily occurs in GSCs and not the more differentiated SGs[13]. Accordingly, genes that regulate rDNA magnification likely have altered activity or expression within GSCs when rDNA CN is low. Thus, we sought to identify genes that are differentially expressed in GSCs under low rDNA CN (magnifying) vs normal rDNA CN conditions. *Drosophila* rDNA loci reside on the sex chromosomes (X and Y), and low rDNA CN conditions can be

generated by combining rDNA-deficient sex chromosomes[5,7]. We used $bb^{Z9}/Ybb^0$ males, which contain reduced X chromosome rDNA CN ($bb^{Z9}$) and no Y chromosome rDNA ($Ybb^0$), for 'low rDNA CN' conditions that induce rDNA magnification[4]. As a 'normal rDNA CN' control, we used $bb^{Z9}/Ybb^+$ males, which contain sufficient rDNA CN on the Y chromosome, and are as genetically matched to $bb^{Z9}/Ybb^0$ males as possible while not inducing rDNA magnification[4]. Each testis contains only ~8–10 GSCs, as opposed to hundreds of more differentiated SGs, spermatocytes, and spermatids (Supplementary Fig. 1A left), therefore making it challenging to isolate enough GSCs to identify GSC-specific transcript changes even using single-cell RNA sequencing (scRNA-seq)[15,16]. To circumvent this challenge, we conducted scRNA-seq in animals with low or normal rDNA CN using testes overexpressing *upd* in their early germline (*nos > upd*). Upd expression in the early germline leads to the overproliferation of GSCs, enriching our cell type of interest (Supplementary Fig. 1A right)[17,18]. Importantly, we confirmed that *upd* overexpression does not interfere with the induction of *R2* expression in GSCs in response to low rDNA CN (Supplementary Fig. 1B-D).

We sought to discover potential regulators of rDNA magnification by identifying genes with altered expression in GSCs with low rDNA CN. To do so, we analyzed the transcriptome of a total of 20,138 quality-verified cells from low and normal rDNA CN *upd*-expressing testes. Cell identity was initially assigned based on rDNA CN condition and previously determined testis cell type expression signatures, revealing cells among all stages of spermatogenesis in our samples (Supplementary Fig. 2A, B, see "Methods")[15,16]. In order to enrich for true positives of differentially-expressed GSC genes, we used two complementary methods to select GSCs for gene expression analysis. In one method, we used iterative sub-clustering to isolate true GSCs, while excluding SG and suspected artifactual GSC:non-GSC cell doublets (Fig. 1C, see "Methods"). The advantage of this cluster-based method is that it included cells with transcriptional signatures similar to GSCs even if they did not express all of the specific markers used to distinguish GSCs. This method resulted in 3087 GSCs to use for differential expression analysis, which identified 721 significantly downregulated and 320 significantly upregulated genes in GSCs with low rDNA CN compared to GSCs with normal rDNA CN (Supplementary Fig. 2C, Supplemental Data 1). The other method identified GSCs based on the expression of specific markers within each cell, regardless of their assigned cluster (Supplementary Fig. 2D, see "Methods"). The advantage of this expression-based method is that it included all cells that may be considered GSCs even if their overall expression profile failed to cluster with other GSCs. This second method produced 746 GSCs for differential expression analysis, and identified 247 significantly downregulated and 167 significantly upregulated genes in GSCs with low rDNA CN (Supplementary Fig. 2E and Supplemental Data 1). Selecting the overlap between these two analyses yielded 202 genes that are downregulated and 117 genes that are upregulated in low rDNA CN conditions in GSCs from both analyses (Fig. 1D, Supplementary Fig. 2F, and Supplemental Data 1). To further narrow down these candidates, we reasoned that regulators of rDNA magnification would have altered expression specifically in GSCs but not SG, since rDNA magnification is a unique feature of GSCs and not the more differentiated SG[13]. Therefore, we assessed SG gene expression for all 319 differentially expressed genes in GSCs, and found that 71 downregulated and 84 upregulated GSC genes have no (or opposite) expression change in SGs under low rDNA CN condition, suggesting these genes change expression only in GSCs in response to rDNA CN (Fig. 1D and Supplemental Data 2).

## rDNA magnification is repressed by insulin-like receptor (InR)

In this study, we focused on candidate negative regulators of rDNA magnification because their functional importance can be assayed relatively easily. We have previously shown that exogenous expression of R2 is sufficient to induce 'ectopic' rDNA magnification[4], and surmised that RNAi-mediated knockdown of negative regulators of rDNA magnification would have a similar effect. Ectopic rDNA magnification is observed in $bb^{Z9}/Ybb^+$ males, which do not normally expand rDNA CN at the $bb^{Z9}$ locus because they have sufficient rDNA CN on the Y chromosome. Ectopic rDNA magnification in these males can be observed by mating them to females harboring the rDNA deletion $Xbb^{158}$ chromosome and assessing the frequency of $bb^{Z9}/Xbb^{158}$ daughters exhibiting wild-type cuticles due to CN expansion at the $bb^{Z9}$ locus (Fig. 2A, see "Methods"). We screened RNAi targeting 19 out of the 71 candidate negative regulators (represented by 28 RNAi lines) for their ability to induce ectopic rDNA magnification when expressed in the germline, based on the availability of RNAi reagents (Supplemental Data 3). We found that an RNAi targeting *Insulin-like receptor* (*InR*) elicited the strongest induction of rDNA magnification (TRiP.JF01482), leading to 34.5% of magnified offspring (Supplemental Data 2 and Fig. 2B). This is in stark contrast to flies with normal rDNA copy number, where magnified offspring are rarely observed. Importantly, there was no difference in the viability of $bb^{Z9}/Xbb^{158}$ daughters between *InR RNAi* expressing fathers and those with no or limited rDNA magnification (Supplementary Fig. 3), indicating that the increase in the observed frequency of magnified offspring is not due to reduced viability of unmagnified offspring. Expression of $InR^{K1409A}$, a dominant negative isoform of InR, in the germline of normal rDNA CN animals ($bb^{Z9}/Ybb^+$; *nos > $InR^{K1409A}$*) also robustly induced ectopic rDNA magnification (Fig. 2B), confirming that the rDNA magnification is caused by loss of InR function instead of off-target effects of RNAi. Importantly, quantification of rDNA copy number by droplet digital PCR (ddPCR) revealed that $bb^{Z9}$ chromosomes inherited from $bb^{Z9}/Ybb^+$; *nos > $InR^{K1409A}$* fathers have on average 16.3 more rDNA copies than $bb^{Z9}$ chromosomes inherited from control males ($bb^{Z9}/Ybb^+$; *nos:Gal4* alone) (191.4 in the offspring from *nos > $InR^{K1409A}$* fathers vs 175.1 in the offspring from control fathers, regardless of their cuticle phenotype; $p < 0.01$, Fig. 2C). Moreover, expression of a constitutively active InR (*nos > $InR^{K414P}$*) substantially reduced the frequency of rDNA magnification in animals with low rDNA CN, which would usually induce strong magnification (Fig. 2B). Since InR is the sole receptor for IIS in *Drosophila* (insulin/insulin-like growth factor signaling)[19], these results raise the possibility that rDNA magnification is controlled by cells' major growth control pathway.

## Expression of the rDNA-specific retrotransposon R2 is repressed by InR

Expression of the rDNA-specific retrotransposon R2 in GSCs is required for rDNA magnification[4]. R2 activity creates DNA breaks at rDNA loci that can be used for USCE to cause rDNA magnification (Fig. 1A)[20]. To determine if the effect of InR to suppress rDNA magnification is via repression of R2 expression, we used RNA FISH to examine whether downregulation of *InR* leads to increased R2 expression in GSCs. While R2 is typically silent in GSCs with normal rDNA CN, we indeed found that inhibition of *InR* via either RNAi (TRiP.JF01482) or expression of the dominant negative $InR^{K1409A}$ isoform increased the portion of GSCs expressing R2, despite having normal rDNA CN (Fig. 2D–H). Conversely, while R2 is typically frequently expressed in GSCs with low rDNA CN, expression of constitutively active $InR^{K1409A}$ reduced the portion of R2 expressing GSCs in this context (Fig. 2H). Furthermore, we found that expression of an RNAi that targets R2[4] suppresses the rDNA magnification caused by the dominant negative $InR^{K1409A}$ (Fig. 2B), revealing that R2 expression is downstream of InR in inducing rDNA magnification. These results indicate that InR represses R2 and downregulation of *InR* in GSCs with low rDNA CN may allow for R2's expression, which in turn induces rDNA magnification.

We recently found that R2 expression in GSCs, but not SGs, is necessary for rDNA magnification, and that R2 is likely inhibited in SGs,

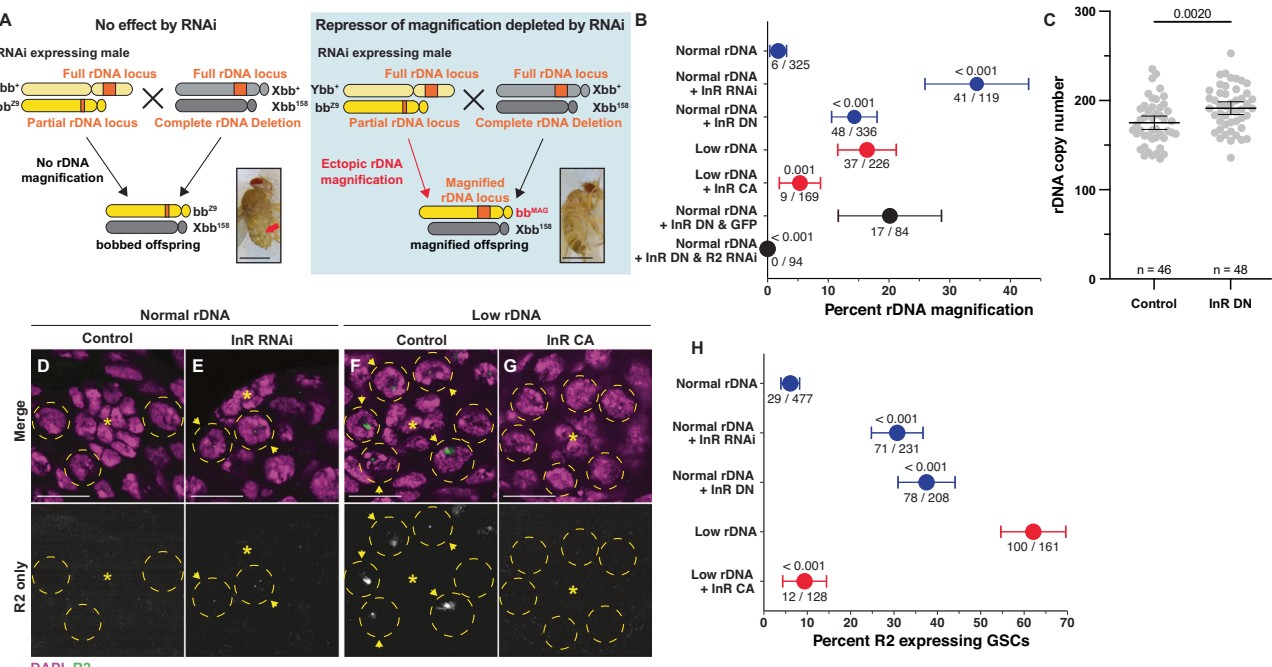

**Fig. 2 | InR represses rDNA magnification and R2 expression. A** Schematic to rapidly assess the function of candidate rDNA magnification repressors. RNAi targeting a candidate gene is expressed in the early germline of males harboring the $bb^{Z9}$ X chromosome with the Y chromosome containing a wild-type rDNA locus ($Ybb^+$). Males are mated to females harboring an X chromosome completely lacking rDNA ($Xbb^{158}$), and the resultant $bb^{Z9}/Xbb^{158}$ daughters are assessed for rDNA magnification based on reversion of the *bobbed* phenotype (indicated by red arrowhead). Scale bar = 1 mm. **B** Frequency of offspring with ectopic rDNA magnification determined by the presence of wild-type cuticle. *p*-value determined by two-sided chi-squared test compared to control condition (Normal rDNA for Normal rDNA + InR RNAi ($p = 2.2 \times 10^{-16}$) and normal rDNA + InR DN ($p = 2.46 \times 10^{-8}$); Low rDNA for Low rDNA + InR CA ($p = 1.25 \times 10^{-3}$); Normal rDNA + InR DN & GFP for Normal rDNA + InR DN and R2 RNAi ($p = 1.49 \times 10^{-5}$)). Data presented are the percent of animals exhibiting rDNA magnification, with source data listed below, and ±95% confidence interval (CI). **C** rDNA CN assessed by ddPCR in individual $bb^{Z9}/Xbb^{158}$ daughters of control of InR dominant-negative expressing males. *p*-value determined by two-tailed Welch's *t*-test. The data presented is a mean value ± 95% CI. **D**–**G** R2 RNA FISH images in normal (**D**, **E**) or low rDNA CN testes (**F**, **G**). DAPI in magenta, R2 in green. Asterisk (*) indicates the hub (stem cell niche), GSCs in yellow dotted circle. R2 positive GSCs indicated by a yellow arrowhead. Scale bar = 10 μm. **H** Frequency of GSCs expressing R2. *p*-value determined by two-sided chi-squared test compared to the control condition (Normal rDNA for Normal rDNA + InR RNAi ($p = 2.2 \times 10^{-16}$) and Normal rDNA + InR DN ($p = 2.2 \times 10^{-16}$); Low rDNA for Low rDNA + InR CA ($p = 2.2 \times 10^{-16}$)). Data presented is the percent of total GSCs observed that express R2, with source data listed below, and ±95% CI. Genotypes: $bb^{Z9}/Ybb^+;;$ *nos-Gal4*/+ (Normal rDNA, Control), $bb^{Z9}/Ybb^+;;$ *nos-Gal4/UAS-InR RNAi^{JF01482}* (Normal rDNA + InR RNAi), $bb^{Z9}/Ybb^+;;$ *nos-Gal4/UAS-InR^{K1409A}* (Normal rDNA + InR DN, InR DN), $bb^{Z9}/Ybb^0;;$ *nos-Gal4*/ + (Low rDNA), $bb^{Z9}/Ybb^0;;$ *nos-Gal4/UAS-InR^{K414P}* (low rDNA + InR CA), $bb^{Z9}/Ybb^+;$ *UAS-InR^{K1409A}*/+; *nos-Gal4/UAS-R2 RNAi-1* (InR DN + R2 RNAi), $bb^{Z9}/Ybb^+;$ *UAS-InR^{K1409A}*/+; *nos-Gal4/UAS-GFP* (Normal rDNA + GFP). Source data are provided as a Source Data file.

since these cells are highly sensitive to DNA damage[4,13,14]. Intriguingly, *InR* expression remains unchanged in SG with low rDNA CN (Supplemental Data 2), suggesting that sustained InR activity in SGs maintains R2 silencing when rDNA CN is low. Interestingly, we find that inhibition of InR activity in SG with normal rDNA CN by *InR^{K1409A}* expression ectopically induces rDNA magnification, albeit at a much lower frequency than InR inhibition in GSCs (Supplementary Fig. 4A). These findings suggest rDNA CN expansion is possible in SGs, but that InR activity suppresses R2 expression in SGs under both normal and low rDNA CN conditions, likely to prevent unnecessary DSBs in SGs that are sensitive to DSBs[14]. The low frequency of magnification in SGs upon reduced InR activity (Supplementary Fig. 4A) is likely due to SGs' inability to undergo repeated rounds of USCE, as in GSCs, and increased SG death caused by R2-induced DSBs[13,14]. These findings indicate that the dynamic regulation of *InR* expression in response to rDNA CN is a unique feature of GSCs that enables rDNA CN expansion to specifically function in these cells. Furthermore, we found that SG that dedifferentiated into GSC (by protein starvation and refeeding[21,22]) (Supplementary Fig. 4B, C) failed to express R2 when rDNA CN is low (Supplementary Fig. 4D). Together, these findings suggest that the mechanism that accomplishes *InR* repression in GSCs under low rDNA CN condition is a unique feature of native GSCs, and that this mechanism to activate R2 expression is irreversibly lost upon commitment to SG state.

## The mechanistic target of rapamycin complex 1 (mTORC1) functions downstream of InR to regulate R2 expression and rDNA magnification

There are multiple different effectors of IIS downstream of InR that can each impact transcriptional, translational, and metabolic activity[23], thus we sought to identify which effector(s) function in InR-mediated repression of R2 and rDNA magnification. One of the major transcriptional effectors of IIS is the transcription factor FoxO, which is phosphorylated through the PI3K/AKT pathway upon IIS activation to sequester FoxO in the cytoplasm and prevent transcription of FoxO targets[24]. We tested whether FoxO transcriptional activity may promote R2 expression and rDNA magnification. However, we found that over-expression of *FoxO* in GSCs did not induce ectopic rDNA magnification in animals with normal rDNA CN, and inhibition of *FoxO* in GSCs through RNAi did not reduce rDNA magnification in animals with low rDNA CN (Supplementary Fig. 5A). We additionally found no difference in FoxO nuclear localization between low and normal rDNA CN GSCs (Supplementary Fig. 5B, C). These results suggest that FoxO is not involved in rDNA magnification downstream of InR. We next tested if PI3K/AKT activity is reduced when rDNA CN is low. We expressed a GFP-tagged pleckstrin homology domain (tGPH), which binds phosphatidylinositol-3,4,5-P$_3$ (PIP$_3$)[25]. PIP$_3$ is formed at the lipid membrane by PI3K during InR stimulation to activate the AKT pathway, and thus membrane tGPH localization indicates active PI3K/AKT signaling[25]. We

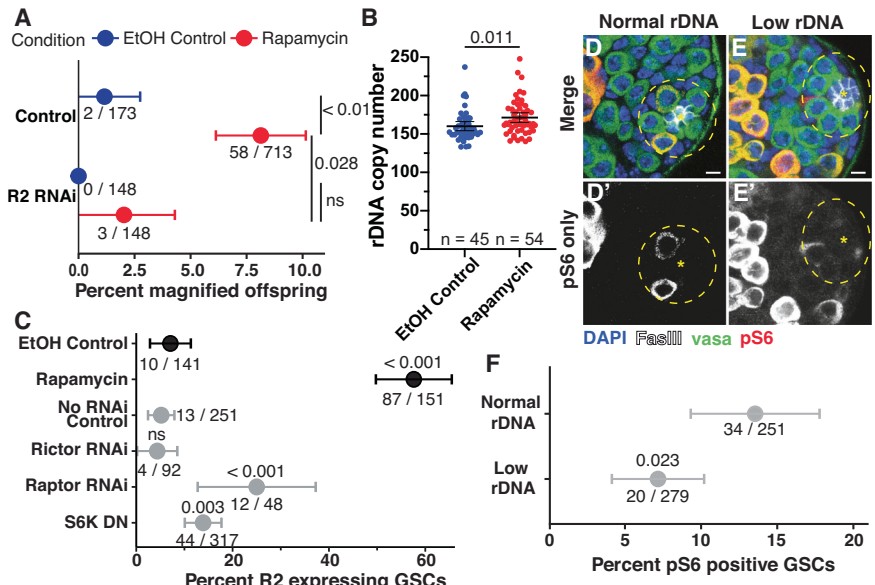

**Fig. 3 | mTORC1 suppresses R2 expression and rDNA magnification. A** Percent wild-type offspring in rDNA magnification assay from 10 μM rapamycin or control ethanol (EtOH) fed males. Animals in both conditions are $bb^{Z9}/Ybb^+;; nos-Gal4/+$. *p*-values are determined by a two-sided chi-squared test. EtOH condition is the control for all comparisons between EtOH and rapamycin treatments ($p = 0.0038$ between control genotype samples; $p = 0.5$ between R2 RNAi samples). Control genotype is the control for comparison between rapamycin-fed treatments ($p = 0.028$). Data presented is the percentage of total animals exhibiting wild-type cuticles, with source data listed below, and ±95% CI. **B** rDNA copy number determined by ddPCR in offspring of 10 μM rapamycin or control EtOH-fed males. Animals in both conditions are $bb^{Z9}/Ybb^+;; nos-Gal4/+$. *p*-values were determined by two-tailed Welch's *t*-test. The data presented is the mean ± 95% CI. **C** Percent R2 positive GSCs in 10 μM rapamycin or EtOH fed control males and inhibition of mTor

factors. *p*-value determined by two-sided chi-squared test for samples to control condition (EtOH for rapamycin treatment ($p = 2.2 \times 10^{-16}$); No RNAi for Rictor RNAi ($p = 1.0$, Raptor RNAi ($p = 6.11 \times 10^{-5}$), and S6K DN ($p = 0.003$)) Data presented is the percent of total GSCs expressing R2, with source data listed below, and ±95% CI. **D**, **E** Images of phosphorylated S6 (pS6) in normal and low rDNA CN testes. All images are to scale. Scale bar: 5 μm. **F** Percent pS6 positive GSCs in low rDNA CN animals compared to normal rDNA CN control. *p*-value determined by a two-sided chi-squared test. Data presented is the percentage of GSCs with pS6 staining, with source data listed below, and ±95% CI. **B**, **E** Genotypes: $bb^{Z9}/Ybb^+;; nos-Gal4/+$ (EtOH control, rapamycin, no RNAi control, and normal rDNA), $bb^{Z9}/Ybb^+;; nos-Gal4/UAS-Rictor RNAi^{HMS01588}$ (Rictor RNAi), $bb^{Z9}/Ybb^+; nos-Gal4/UAS-Raptor RNAi^{JF01088}$ (Raptor RNAi), $bb^{Z9}/Ybb^+; UAS-S6k^{K109Q}/+; nos-Gal4/+$ (S6K DN), $bb^{Z9}/Ybb^0; nos-Gal4/+$ (Low rDNA). Source data are provided as a Source Data file.

found no difference in GSC tGPH membrane localization in magnifying compared to non-magnifying conditions (Supplementary Fig. 5D, E), indicating PI3K activity is not reduced during rDNA magnification. Furthermore, we found that RNAi-mediated inhibition of the *Pdk1* kinase, the major mediator of $PIP_3$ that activates AKT[26], does not induce ectopic rDNA magnification (Supplementary Fig. 5A). Together, these results indicate that neither FoxO-dependent nor -independent PI3K/AKT activity functions to repress rDNA magnification.

Having established that PI3K/Akt is unlikely the mediator of repressing rDNA magnification downstream of IIS, we tested the involvement of mTOR. mTOR is a downstream effector of IIS that regulates transcriptional activity and can be stimulated in both InR-dependent and -independent manner[27]. We found that mTOR inhibition through rapamycin feeding induced ectopic rDNA magnification in males with normal rDNA CN (Fig. 3A). Quantification of rDNA CN by ddPCR revealed that the offspring of rapamycin-fed males inherited 11.1 more rDNA copies than the offspring of control males (mean rDNA copy number of 160.2 in the offspring of ethanol only controls and 171.3 in rapamycin fed males), regardless of cuticle phenotype (Fig. 3B). Rapamycin feeding similarly increased the portion of R2 expressing GSCs in normal rDNA CN males (Fig. 3C). In addition, RNAi-mediated knockdown of R2 suppressed rDNA magnification caused by rapamycin (Fig. 3A), indicating that mTOR represses rDNA magnification via the suppression of R2. mTor functions in two complexes, mTORC1, and mTORC2[28]. We found that RNAi-mediated inhibition of the mTORC1-specific factor *Raptor* increased the portion of R2 expressing GSCs, but there was no effect of inhibiting the mTORC2-specific factor *Rictor* (Fig. 3C). We found that expression of a dominant negative

isoform of S6 kinase ($S6k^{K109Q}$), a major target of mTORC1[29], also increased the portion of R2 expressing GSCs (Fig. 3C). Furthermore, we found that the portion of GSCs with phosphorylated ribosomal protein S6, a readout of mTORC1 activity[30,31], is reduced in animals with low rDNA CN (Fig. 3D–F) indicating that mTORC1 activity is reduced in GSCs in response to low rDNA CN. Together these results indicate that mTORC1 represses R2 expression when rDNA CN is abundant, but mTORC1 activity is reduced when rDNA CN is low, allowing for R2 expression and the induction of rDNA magnification.

## Dietary condition influences germline R2 expression and rDNA magnification activity

Because IIS and mTor are central mediators of nutrient signaling[23], we postulated that rDNA magnification might be influenced by nutrient conditions via these pathways. IIS and mTor stimulation by high caloric diets has been shown to reduce inherited rDNA CN[32] and we reasoned that nutrient conditions may dynamically alter rDNA magnification activity. To test this notion, we examined if changes in dietary conditions influenced R2 expression similar to modulation of InR and mTor activity. Males were fed for 5 days on varying diets containing the same amount of dietary sugar (5% sucrose), but with modified nutritional yeast (1%, 5%, and 30%). We respectively call these diets SY1, SY5, and SY30, with the SY5 diet most closely matching the nutritional content of the standard food used in our other experiments. We found that, while animals with normal rDNA CN fed on standard food (and SY5 food) rarely express R2 (only 5% of GSCs), the same animals fed on SY1 food had an increased frequency of R2 expressing GSCs (Fig. 4A, B, and E). Conversely, animals with

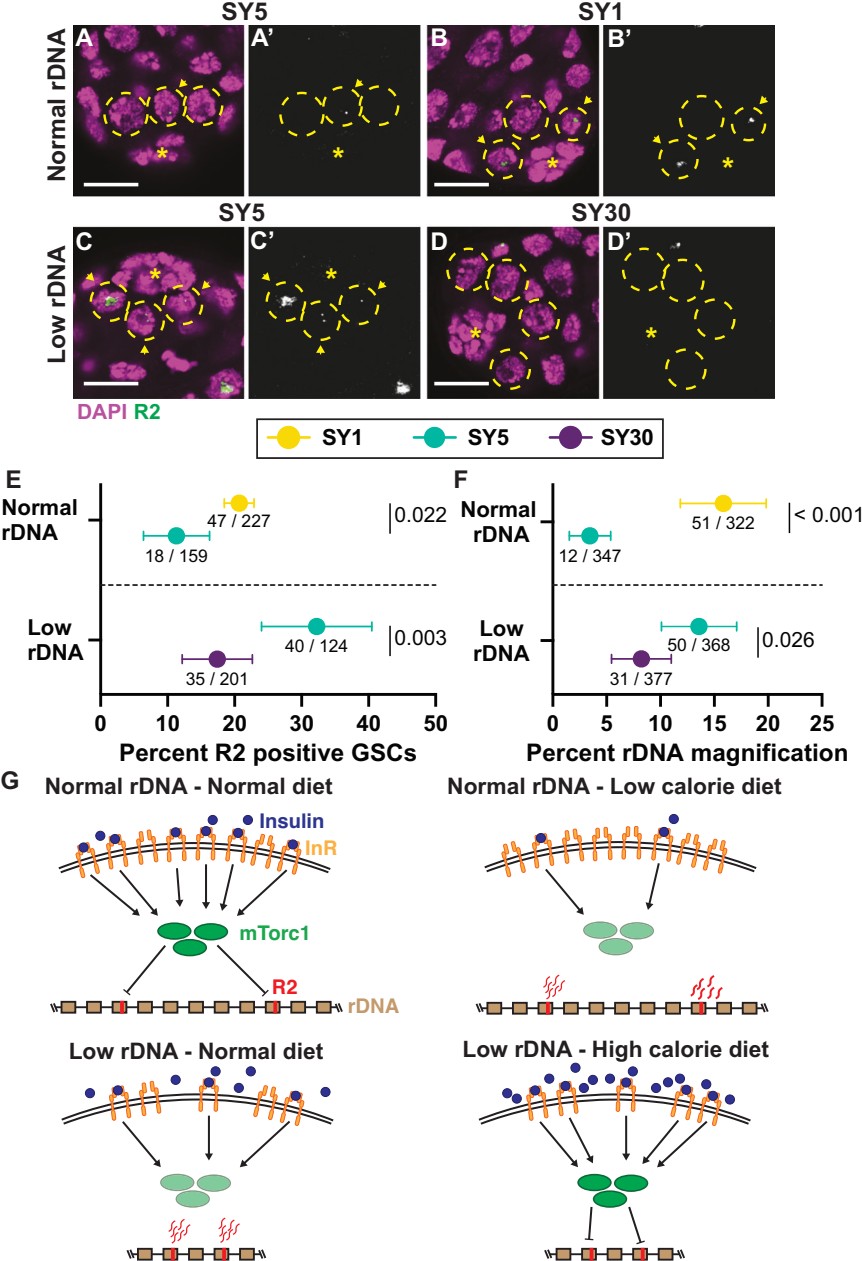

**Fig. 4 | Dietary conditions alter germline R2 expression and rDNA magnification activity. A–D** Images of R2 FISH in testes of low and normal rDNA CN. DAPI in magenta and R2 in green. **A'–D'** is the R2 channel only. All images are to scale. Scale bar: 10 μm. **E** Percent R2 positive GSCs in males with low or normal rDNA CN on low and high-calorie diets. **F** Percent offspring with wild-type cuticles in rDNA magnification assay of from males with normal or low rDNA CN fed on low and high-calorie diets. For **E**, **F**, data presented is the percentage of total GSCs expressing R2 (**E**) or total animals exhibiting wild-type cuticles (**F**), with source data listed below, and ±95% CI. The *p*-value is determined by a two-sided chi-squared test, and the SY5 diet is the control condition for all experiments. $p = 9.01 \times 10^{-8}$ for percent rDNA magnification between normal rDNA SY1 and SY5 **G** Model of dietary impact on rDNA copy number regulation through InR. Under conditions of normal rDNA copy number and normal diet (top left), activated insulin receptors in GSCs stimulate mTORC1 activity, which represses R2 expression. Low-calorie diets (top right) reduce insulin signaling, limiting mTORC1 stimulation, relieving R2 repression, and allowing for R2 to stimulate rDNA copy number expansion. When rDNA copy number is low under normal conditions (bottom left), reduced expression of the insulin receptor in GSCs also reduces mTORC1 activation, leading to R2 expression that stimulates rDNA copy number expansion. When high-calorie diets increase insulin stimulation in low rDNA GSCs (bottom right), the increased insulin signaling activates mTORC1 sufficiently to suppress R2 expression and limit rDNA copy number expansion activity. Genotypes: *bb^Z9^/Ybb^+^;; nos-Gal4/* + (Normal rDNA), *bb^Z9^/Ybb^0^;; nos-Gal4/* + (low rDNA). Source data are provided as a Source Data file.

low rDNA CN on SY5 food exhibit a high frequency of R2-expressing GSCs but feeding them on SY30 diets reduced the frequency in R2 expressing GSCs (Fig. 4C–E). These results indicate that R2 expression in GSCs is regulated in response to nutritional inputs, in a manner that induces R2 under lower nutrient conditions but suppresses its expression under high nutrient conditions.

Moreover, we found that these dietary conditions influence rDNA CN expansion activity, suggesting that R2 expression controlled by dietary conditions is functionally linked to rDNA CN changes. Normal rDNA CN males were raised on SY1 or SY5 media for their first 10 days of adulthood, then mated to females on standard food to assess the frequency of rDNA magnification among their offspring (See

"Methods", Supplementary Fig. 6A). We observed that rDNA magnification was very rare from males raised on SY5 media, but that rDNA magnification was dramatically increased in males raised on SY1 media (Fig. 4F). We also found that the offspring of males fed SY1 media inherited 14.3 more rDNA copies than the offspring of males fed SY5 media, irrespective of whether they recovered from bobbed phenotype or not (determined by ddPCR, $p = 0.0315$) (Supplementary Fig. 6B). In contrast, animals with low rDNA CN, which would normally induce magnification, had reduced rDNA magnification when fed with SY30 food for 10 days (Fig. 4F). Offspring of low rDNA CN males fed SY30 diet also had 12.1 fewer rDNA copies than offspring of low rDNA CN males on the SY5 diet ($p = 0.0463$) (Supplementary Fig. 6C). Together, these results indicate that this high caloric diet suppresses rDNA magnification. The low rDNA CN males raised on SY30 food still experienced some rDNA magnification though, indicating that high caloric diets dampen, but do not completely suppress, rDNA magnification activity. Together, these findings reveal that dietary inputs influence R2 regulation, and in turn rDNA magnification activity, likely by impacting the IIS/mTOR pathways.

## Discussion

rDNA loci are essential but inherently unstable genetic elements, and their maintenance is particularly critical in the germline lineage for their continuation through generations. We previously showed that the rDNA-specific retrotransposon R2 plays a critical role in inducing rDNA magnification in *Drosophila* GSCs, representing a rare example of host-TE mutualism[4], instead of them being 'genomic parasites' as generally regarded. However, for such a mutualistic host-TE relationship to work, TEs' activity must be precisely regulated to limit TE expression when most beneficial, while preventing unnecessary transposition that would threaten host genome integrity. That is, R2's expression/activity must be integrated within the host's signaling network to allow a mutually beneficial relationship.

In this study, we investigated such a mechanism by which R2 regulation is integrated into the host's physiological system to sense rDNA CN. We used scRNA-seq to identify host factors that are differentially expressed in GSCs during rDNA magnification. Through this approach, we found that InR is a major negative regulator of rDNA magnification. *InR* transcripts were downregulated in response to low rDNA CN, and this downregulation was necessary to induce R2 expression and rDNA magnification. Moreover, we found that mTORC1 is the downstream effector of InR regulating R2 expression: mTORC1 is required to silence R2 expression when the animals have sufficient rDNA CN, whereas mTORC1 activity is downregulated in GSCs with low rDNA CN, which leads to R2 derepression and rDNA magnification. These results reveal that regulation over rDNA CN expansion and R2 expression is deeply integrated with the host's major nutrients sensing pathway (Fig. 4G).

Intriguingly, we found that dietary conditions, which influence IIS and mTORC1 activity, can alter R2 expression and rDNA magnification activity. These results have an important implication: not only that animals (and GSCs within) have the ability to sense rDNA CN to regulate rDNA magnification, but that the number of rDNA copies may also be under the influence of nutrient conditions. That is, even when animals have low rDNA CN that would normally induce magnification, high nutrients dampened rDNA magnification. On the contrary, low nutrient conditions induced rDNA magnification even if animals had normal rDNA CN. Our findings suggest that environmentally influenced insulin activity is an endogenous feature of rDNA CN regulation, though it remains unclear why this influence exists. It is possible that dietary conditions alter the physiologically required number of rDNA copies or potentially distort the physiological readout of rDNA CN. Regardless of the source of diet-induced rDNA CN changes, these findings highlight a counterintuitive relationship between nutrients and rDNA maintenance: rDNA CN is not maintained in conditions when

rRNA synthesis is most needed, but rDNA CN is increased when biosynthetic resources are limited. Although counterintuitive, this aligns with the known relationship between nutrients and rDNA CN. For example, high-nutrient diets have been shown to cause somatic and germline rDNA CN reduction in *Drosophila* in an IIS- and mTor-dependent manner[32]. Furthermore, increased mTor activity was shown to cause rDNA CN reduction in mouse hematopoietic stem cells, and some cancer types with high mTor activity are associated with low rDNA CN[33]. It remains unclear as to why these conditions with high biosynthetic activity are associated with reduced rDNA CN that may hinder ribosome biogenesis. It was proposed that reduced rDNA CN may be selected in highly mitotic cells to help expedite DNA replication, particularly in conditions of replication stress[33,34]. GSC proliferation rate is associated with nutrient availability[35], suggesting IIS and mTor may similarly optimize rDNA CN for replicative demand in GSCs. Alternatively, these observations may indicate that the mechanisms of rDNA CN expansion are not compatible with rRNA transcription, preventing rDNA CN expansion when rRNA synthesis is most active. Indeed, replication-transcription conflict is a source of CN destabilizing DNA breaks at rDNA loci in many organisms[36,37], suggesting a similar conflict between replication and magnification may also destabilize rDNA CN. Counter to our observations in *Drosophila*, rDNA CN expansion in yeast is actually stimulated by mTor and suppressed by low-calorie conditions[38]. The differences between yeast and flies that underlie the opposing effects of nutrition on rDNA CN regulation remain unclear. Further investigation into the specific mechanisms of rDNA CN regulation, as well as the specific physiological consequences of high and low rDNA CN in multicellular organisms, is needed to understand why rDNA CN and nutrient status sensing are integrated to dynamically regulate rDNA CN expansion.

How IIS and mTor regulate R2 expression to control rDNA magnification awaits future investigation. R2 lacks its own promoter, therefore its expression is entirely dependent on read-through transcription of the rDNA copy where it is inserted[39]. mTORC1 is a known positive regulator of rDNA transcription via its interaction with Pol I recruitment factors[40,41], yet counterintuitively our data suggests that mTORC1 activity suppresses transcription at R2 containing rDNA copies. It is possible that mTORC1 differentially regulates R2-inserted vs -uninserted rDNA copies, promoting R2-uninserted rDNA copies, while negatively regulating R2-inserted rDNA copies. Indeed, mTor activity is known to increase total rRNA transcription without increasing R2 expression[32]. Further investigation into the mechanisms of rDNA transcriptional regulation and the effects of mTORC1 activity on the transcription of R2-containing rDNA copies is needed to fully understand how R2 is regulated in response to rDNA CN. Interestingly, IIS and mTor activity are also well-described regulators of lifespan across eukaryotes, and their inhibition by dietary or genetic manipulation extends lifespan[42], although the mechanisms that mediate these effects remain unclear[43]. The instability of rDNA has been proposed to be a major factor contributing to replicative senescence in yeast[44], and yeast mutants that increase or reduce rDNA CN stability, respectively lengthen and shorten replicative lifespan[45–47]. The possibility that IIS and mTOR inhibition may allow for rDNA CN expansion during aging suggests that rDNA CN preservation may be a mechanism for their inhibition to extend lifespan.

The nature of the mechanism that senses rDNA CN in the *Drosophila* germline has remained elusive since rDNA magnification was discovered over 50 years ago. The present study provides insight into how the CN sensing might be transmitted to the mechanism that induces rDNA magnification. Further investigation of the other candidates found in this study may provide further insights into the mechanism of rDNA CN sensing. Conditions that disrupt ribosome biogenesis or rRNA synthesis have been shown to lead to increased R2 expression[48,49], and we found that RNAi that target three different ribosomal proteins induced rDNA magnification (Supplemental Data 3,

*RpL10[TRiP.JF02520]*, *RpL15Aa[TRiP.HMS00968]*, and *RpS5a[TRiP.GL01502]*), suggesting reduced ribosome biogenesis or function may signal GSC rDNA insufficiency. It remains unclear if ribosome biogenesis or function is actually disrupted in the germline of bobbed animals: however, we found over a quarter of genes encoding for ribosomal proteins have reduced expression in GSCs with low rDNA CN (Supplemental Data 1, 22.9-fold enrichment, $p < 10^{-15}$ determined by two-sided chi-squared test). This enrichment for ribosomal proteins to have reduced expression implies that ribosome activity is indeed reduced in low rDNA CN condition and may serve as a sensor of reduced rDNA CN, although ribosome activity alone does not explain why Y chromosome rDNA loss, but not X chromosome rDNA reduction, is necessary to induce rDNA magnification, as observed in earlier studies[6]. The involvement of IIS and mTOR in regulating rDNA magnification suggests low rDNA CN may be sensed by these pathways through changes in metabolic conditions, such as glucose or amino acid availability[23]. Indeed, many ribosome deficiencies are associated with altered glycolysis, amino acid synthesis, and proteasome activity[50], suggesting metabolism may become altered when rDNA CN is reduced. Our finding that dietary manipulation impacts R2 expression and rDNA CN expansion activity further suggests a key role for metabolite availability in rDNA CN sensing. Alternatively, stabilized p53 is a common consequence of disrupted ribosome biogenesis, including in *Drosophila* female GSCs[51,52], which may repress insulin and mTor activity in GSCs with reduced rDNA CN. Intriguingly, our findings suggest that the inability of SG to induce R2 expression when rDNA CN is low is due to an inability to reduce *InR* expression upon rDNA CN loss, suggesting GSCs have unique mechanisms to sense or signal reduced rDNA CN. Investigation into the differences in insulin signaling regulation between GSCs and SG may reveal the mechanisms that sense and signal low rDNA conditions in GSCs. Uncovering how rDNA CN is sensed by GSCs and the potential role of metabolites in this function is critical to understanding the physiological consequences of the integrated sensing of environmental and genomic conditions to regulate rDNA CN.

Taken together, the present work demonstrates that repression of the rDNA-specific retrotransposon R2 by the IIS/mTor pathway regulates rDNA maintenance activity in GSCs. We propose that this role for IIS in rDNA regulation integrates nutrient and rDNA CN regulation to transgenerationally maintain and adjust rDNA CN. Such a dynamic mechanism may explain the widespread intra-species variation in rDNA CN observed in many organisms[53–56]. Excessive mTor activation has been associated with some instances of rDNA CN reduction in mice and humans[33], suggesting its function to regulate rDNA CN may be widely conserved across animals. These observations may reflect a conserved nature of rDNA CN dynamics and regulation to integrate natural CN fluctuation with changing environmental factors such as nutrients.

## Methods

### Single-cell RNA sequencing sample preparation
50 testes were hand dissected from 1 to 5-day-old flies in 1× PBS and transferred immediately into tubes with cold 1× PBS on ice. Tissue dissociation was performed as previously described[16]. Cell viability and density were determined on a hemocytometer using Trypan Blue stain and DIC imaging. Cells were processed for library preparation using the 10× genomics chromium controller and chromium single-cell library and gel bead kit following standard manufacturer's protocol. Amplified cDNA libraries were quantified by bioanalyzer, size selected by AMPure beads, and sequenced on a NovaSeq SP.

### Single-cell sequencing data analysis
The 10× cell ranger (v7.1.0) pipeline was used with default parameters to map reads to the DM3 reference genome. The resulting matrices were read and processed into a single combined data set using the

Seurat R package (v4)[57]. Cells containing between 5500 and 250,000 counts and 200 and 9000 features were isolated for data analysis to enrich for analysis of intact isolated cells. The relative similarity and differences in gene expression for all 20,138 combined cells from low rDNA CN and normal rDNA CN nos > upd samples were reduced using 14 dimensions and a resolution of 0.5 in a Uniform Manifold Approximation and Projection (UMAP)-based dimensionality reduction for cluster analysis. Previously determined expression signatures for distinct spermatogenesis developmental stages and testis cell types were used to initially assign cell type designations to the UMAP clusters[16]. Specifically, clusters that predominantly contained *vas*, *nos*, and *ovo* expressing cells were considered GSC and SG; *dlg1*, *CadN*, *tj*, and *zfh1* labeled Cyst cell clusters; *fzo* and *CycB* labeled spermatocytes; and *CG32106* and *m-cup* identified spermatids. Any unlabeled clusters were subsequently categorized based on the cell type associated with their strongest unique expression markers in the previous analysis[16].

GSCs were determined for differential expression analysis based on sub-clustering and expression selection methods. Sub-clustering renormalized data from cells in GSC/SG containing clusters and a Cyst cell cluster outgroup and gene expression differences were again reduced using 14 dimensions for UMAP-based dimensionality reduction. GSCs containing clusters were distinguished from SG based on the high concentration of cells expressing *nos*, *vas*, and *ovo*, while SG-containing clusters were *vas*-positive, but mostly devoid of *nos* and *ovo* expression. Cyst cell-containing clusters were again identified by *dlg1*, *CadN*, *tj*, and *zfh1*, and clusters containing both GSC and Cyst cell makers were considered a mixed population cluster and excluded from the analysis. For expression selection methods, cells from any original cluster were isolated based on positive expression for *vas*, *nos*, and *ovo*, but negative expression for *tj* and *zfh1*. Genes with a $\log_2$ fold-change greater than 0.25 and a Bonferroni corrected *p*-value less than 0.05 were considered significantly differentially expressed. Potential false-positive expression increases due to contaminating ambient RNA from lysed cells were eliminated by removing all genes with increased expression both low rDNA CN somatic cells and GSCs.

### DNA isolation and rDNA copy number measurement by droplet digital PCR (ddPCR)
DNA was isolated from individual *Drosophila* adults using a modified DNeasy Blood and Tissue DNA extraction kit (Qiagen), as previously described[4]. In short, animals were homogenized in 200 μL Buffer ATL containing proteinase K and vortexed for 15 s. Samples were then incubated for 1.5 h at 56 °C and prepared following the manufacturer's protocol. DNA concentration and purity of all samples were determined by NanoDrop One spectrophotometer (ThermoFisher).

rDNA copy number quantification was determined by ddPCR by assessing 28S copy number normalized to control genes (*RpL49* and *Upf1*) as previously described[4]. In short, control gene and 28S analysis were performed in two separate reactions, with 30 ng of DNA sample used per 20 μL ddPCR control gene reaction, and 0.3 ng of DNA per 20 μL ddPCR 28S rDNA reaction. 28S reactions included 4 U HindIII-HF restriction enzyme (New England Biolabs) and incubated at room temperature for 15 min prior to droplet generation. ddPCR droplets were generated from samples using QX200 Droplet Generator (Bio-Rad) with ddPCR Supermix for Probes (no dUTP) (Bio-Rad) according to the manufacturer's protocol. PCR cycling was completed on a C100 deep-well thermocycler (Bio-Rad) and fluorescence was measured by the QX200 Droplet Reader (Bio-Rad) and the sample copy number was calculated using Quantasoft software (Bio-Rad). rDNA copy number per genome was determined by 28S sample copy number multiplied by 100 (due to the 100× dilution of the sample in the 28S reaction compared to the control reaction) divided by 2× control gene copy number. The 28S copy number normalized to each control gene was averaged to determine the 28S copy number for each sample. Primers and probes are listed in Supplemental Data 4.

### rDNA magnification assay

For rDNA magnification assays, $Ybb^0/bb^{Z9}$ males were used for 'low rDNA' conditions, and $Y/bb^{Z9}$ males were used for 'normal rDNA' conditions. To assess the frequency of rDNA magnification, males were mated in bulk to $bb^{158}/FM6$, $Bar$ females. The result in $bb^{Z9}/bb^{158}$ female offspring was selected based on the absence of the $Bar$ dominant marker and scored for the presence of the $bobbed$ phenotype. The portion of offspring having wild-type cuticles and not the $bobbed$ phenotype represents the frequency of rDNA magnification. Wild-type and $bobbed$ example images were taken using a View4K Microscope Camera (Microscope Central) on a stereomicroscope and the Toup-View software (Hangzhou ToupTek Photonics Co.).

### RNA FISH

RNA FISH samples were prepared using an R2 Stellaris probe set (Biosearch Technologies) as previously described[3]. Dissected testes were fixed for 30 min in 4% formaldehyde in PBS, briefly washed twice in PBS for 5 min, and permeabilized in 70% ethanol at 4 °C overnight. Samples were briefly washed in 2× saline-sodium citrate (SSC) with 10% formamide, then hybridized with 50 nM probes at 37 °C overnight. Following hybridization, samples were washed twice with 2× SSC containing 10% formamide for 30 min and mounted in VECTASHIELD with DAPI (Vector Labs). Samples were imaged using a Leica Stellaris 8 confocal microscope with a 63× oil-immersion objective and processed using Fiji (ImageJ) software.

### Immunofluorescence

Immunofluorescence staining of testes was performed as previously described[58]. Testes were dissected in 1× PBS and fixed in 4% formaldehyde in PBS for 30 min. Following fixing, testes were briefly washed two times in 1× PBS containing 0.1% Triton-X (PBS-T), followed by a 30-min wash in PBS-T. Samples were subsequently incubated at 4 °C overnight with primary antibody in 3% bovine serum albumin (BSA) in PBS-T. Samples were washed three times for 20 min in PBS-T, then incubated with secondary antibody in 3% BSA in PBS-T at 4 °C overnight. Samples were then washed three times again for 20 min in PBS-T and mounted in VECTASHIELD with DAPI (Vector Labs). Primary antibodies used in this study can be found in Supplemental Data 4. Images were taken with a Leica Stellaris 8 confocal microscope with 63× oil-immersion objectives and processed using Fiji (ImageJ) software.

### *Drosophila* genetics and dietary conditions

*Drosophila* lines used in this study can be found in Supplemental Data 4. All animals were reared at 25 °C on standard Bloomington medium, except when specific diets were mentioned. Zero to five-day-old adults were used for all experiments, except when specific ages were mentioned. Experimental diets consisted of 1% agar, 5% sucrose, and respectively 1%, 5%, or 30% yeast for SY1, SY5, and SY30 diets. Animals were raised on standard medium, and newly eclosed adult males were transferred to experimental diets for 5 days prior to dissections for RNA FISH, or 10 days prior to magnification assay.

Complete protein starvation and GSC dedifferentiation paradigm were used as previously described[21]. In short, less than 24-h old males were transferred to either standard food (fed), or 16% sucrose/0.7% agar food (starved) in groups of 20–40 flies per vial. Animals were given new food every 3–4 days. After 21 days, starved animals were reintroduced to standard food.

### Rapamycin feeding

Ten-micrometer rapamycin food was prepared by adding rapamycin dissolved in ethanol directly to vials containing standard corn-meal-based food. Control food was prepared by adding an equal volume of ethanol. Fifteen newly eclosed adult males were placed in vials and raised at 25 °C for 5 days. After the feeding course, males were mated to females on standard food for magnification assays.

### Statistics

Statistical significance was determined by a two-tailed chi-squared test for all comparisons of the percentage of samples with categorical values (percent magnification or R2 positive cells), and error bars were generated using the Confidence Interval for a Population Proportion formula. For all comparisons of samples with independent values (rDNA copy number), significance was determined by two-tailed Welch's $t$-test, and the error represents a 95% confidence interval. The significance of differential gene expression analysis was determined by a non-parametric Wilcoxon rank sum test. All represented $p$-values are Bonferroni corrected based on the number of comparisons. All measurements were taken from distinct samples.

### Inclusion and ethics statement

The authors affirm that this research was conducted with a commitment to inclusivity and ethical considerations, including adhering to all relevant local regulations of environmental protection.

### Reporting summary

Further information on research design is available in the Nature Portfolio Reporting Summary linked to this article.

## Data availability

The sequencing datasets have been deposited in NCBI's Gene Expression Omnibus and are accessible through GEO Series accession number GSE263351. Source data are provided in this paper.

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

## Acknowledgements

We thank the Bloomington Drosophila Stock Center, Kyoto Drosophila Stock Center, FlyBase, and Developmental Studies Hybridoma Bank for reagents and resources. We thank Dr. Aurelio Teleman and Dr. Jong-kyeong Chung for sharing reagents, and Dr. Joshua Dubnau for sharing equipment. We thank the Yamashita lab members for their discussion and comments on the manuscript. We also thank the Whitehead Institute Genome Technology Core for their assistance with scRNA-seq. This research was supported by NIH F32GM143850 (A.A.R), the Howard Hughes Medical Institute (Y.M.Y.), the John Templeton Foundation (Y.M.Y) , and start-up funds provided by the Stony Brook University Department of Biochemistry and Cell Biology and the Renaissance School of Medicine (J.O.N.).

## Author contributions

J.O.N.: conceptualization, design, data acquisition, data analysis, data curation, writing—original draft, writing—review and editing, data visualization, and funding acquisition. A.S.: data acquisition, data analysis, data curation, and writing—review and editing. A.A.R.: data acquisition, data analysis, writing—original draft, writing—review and editing, and funding acquisition. Y.M.Y.: conceptualization, design, data analysis, writing—original draft, writing—review and editing, data visualization, project administration, and funding acquisition.

## Competing interests

The authors declare no competing interests.
