## [Transparent Peer Review file · Nature Communications]

Insulin signaling regulates R2 retrotransposon expression to orchestrate transgenerational rDNA copy number maintenance

Corresponding Author: Dr Jonathan Nelson

Version 0:

Reviewer comments:

Reviewer #1

(Remarks to the Author)

The manuscript "Insulin signaling regulates R2 retrotransposon expression to orchestrate transgenerational rDNA copy number maintenance" describes single cell analysis and genetic analysis implicating Insulin signaling in controlling rDNA magnification in the *Drosophila* male germline. This is an interesting paper that demonstrates new molecular connections between a classical chromatin biology phenomenon and metabolic control.

The presentation of the data raises some questions which should be addressed in the text. The authors use RNAi to test candidates for effects on magnification, but mainly discuss only one - the Insulin Receptor. However in the supplementary table, four lines targeting InR were tested, and only one increased magnification. These others should be mentioned and explained. Is it possible that the one line that works has off-target effects? Additionally, the table lists 3 ribosomal proteins as substantially increasing magnification. This seems worthy of comments in the main text, as well as the other 6 factors that have substantial effects.

The figure legends and labeling should be expanded with more detail. The figures refer to "normal rDNA", "Low rDNA", "normal rDNA + InR RNAi", "low rDNA + InR CA". These should be given as genotypes, and explained in the legends. Figures 3A, B, E should be more clearly labeled. Figure 4E and F seem to be missing a sample each.

Reviewer #2

(Remarks to the Author)

In this compelling manuscript, Nelson et al extend upon their previous work identifying a critical role for the R2 transposon in regulating rDNA copy number to connect R2 regulation with upstream signaling pathways and physiological responses to nutrient sensation. The authors use a combination of RNA sequencing and genetic approaches to identify InR as a critical regulator of rDNA CN. Thorough analysis of potential downstream pathways identified mTOR1 as the proximate regulator of R2 expression upon InR inhibition. Finally, the authors find that dietary conditions, known to influence InR activity, ALSO controls R2 expression and regulation of rDNA CN. This is a high impact, exceptionally well-written manuscript with rigorous, clear results. Upon addressing some minor issues, I am enthusiastic about publication of this manuscript.

- The authors beautifully show the pathway from InR inhibition due to as yet unidentified mechanisms induced by low rDNA CN, through mTOR1 and effects on R2 expression. In addition, they clearly show effects of dietary conditions on regulation of R2 expression independent of the rDNA CN state of GSCs.

For this reviewer, what is missing are experiments related to the consequence of this dietary regulation over rDNA CN. There are several known consequences to GSCs and early spermatogonia due to changes in nutrition, including work from the Yamashita lab showing decreased GSCs due to starvation followed by de-differentiation to restore the stem cell pool following refeeding. It would be interesting to know whether forced depletion of R2 under starvation conditions alters the ability of the testis to appropriately restore homeostasis upon refeeding. Likewise, does inhibition of InR and/or expression of R2 under well-fed conditions cause a phenotypic consequence? Additional data parsing the degree of overlap between the

“endogenous” vs “dietary” responses impacting rDNA CN and the consequences of altering the impact of dietary regulation on the “endogenous” rDNA CN pathway would add significantly to the impact of this work.

- This paper describes complicated interplays between signaling pathways and dietary conditions which is handled quite clearly in the text. One place where the outcomes of the work could be more clearly depicted is in the model shown in Fig.4. While the circular model showing connections between pathway components is compelling, including a more straight-forward and explicit set of diagrams to explain the impact of the work would be helpful. Specifically, showing the explicit outcomes of: 1. Normal food, effect of low rDNA; 2. Low protein food, high rDNA / low rDNA; 3. High protein food, high rDNA / low rDNA in terms of effects on InR/mTOR1 activity, R2 expression and changes (or not) in rDNA CN would be very helpful. And performing experiments suggested above would permit the authors to also include a phenotypic, tissue-level outcome for each of these conditions which would be quite compelling.

Reviewer #3

(Remarks to the Author)

In the manuscript entitled “Insulin signaling regulates R2 retrotransposon expression to orchestrate transgenerational rDNA copy number maintenance”, Nelson and colleagues explored the role of Insulin-mTORC1 axis in regulating ribosomal DNA (rDNA) copy number (CN) in *Drosophila* male germline stem cells. The authors utilized the single-cell RNA sequencing and single molecular RNA FISH to show that insulin signaling and mTOR repress the activity of rDNA-specific R2 retrotransposon, which in turn affect rDNA magnification. While the functions of R2 and Insulin/mTOR in rDNA magnification have been previously reported (PMC4401788 and PMC10266012), the current study’s contribution to the new knowledge appears limited. Thus, it would not be recommended to be published on Nature Communications.

Major

- 1.The author used the bobbed score of offspring to quantify the rDNA magnification. There is a concern about potential overestimation of rDNA magnification due to a competitive advantage of normal rDNA CN sperms over low CN ones. It would indeed be beneficial if the authors could quantify the number of offspring produced by males with low vs. normal rDNA CN.
- 2.In Figure 2B and 2H, the penetrations of rDNA magnification and R2 expression in GSCs are not consistent, especially for the Normal rDNA + InR RNAi and Low rDNA groups (similarly in the Figure 3A,B, Rapamycin group). The author should clarify why a higher percentage of R2 expressing in low rDNA group leads to a lower percentage of rDNA magnification.
- 3.To directly demonstrate that the InR pathway represses rDNA magnification through R2, the author should conduct a rescue experiment where R2 RNAi is used to suppress rDNA magnification in the normal rDNA + InR RNAi condition.
- 4.Since nos-GAL4 is not exclusively expressed in germline stem cells (GSCs), using bam-GAL4 would be crucial for excluding the confounding effects from differentiated germ line cells to draw the conclusion that InR functions in GSCs.
- 5.Providing comprehensive rDNA CN data beyond the bobbed score is essential for a full understanding of the rDNA magnification being studied. It would be better if the authors can provide all additional rDNA CN data besides the bobbed score as shown in Figure 2C.

Minor

- 1.All the *Drosophila* should be italicized.
- 2.Pi3K should be PI3K.

Version 1:

Reviewer comments:

Reviewer #1

(Remarks to the Author)

The text added in the revised manuscript has addressed my concerns.

Reviewer #2

(Remarks to the Author)

The authors have handled all of my original concerns.

Reviewer #3

(Remarks to the Author)

Thank you to the authors for the detailed response and clarification. I appreciate the additional insights into how the present work provides a reinterpretation of the previous study. I have no further questions.

We thank the reviewers for their thoughtful comments that have improved the manuscript. We have incorporated their proposed changes and included additional experiments to address their comments (**Fig 2B, 3B, 4E-G, S3, S4, and S6B-C**). Comments from each reviewer are addressed in the following point-by-point responses. All changes are highlighted in the manuscript. We hope that you will find that these revisions and comments have addressed all concerns raised by reviewers.

Reviewer #1 (Remarks to the Author):

The manuscript “Insulin signaling regulates R2 retrotransposon expression to orchestrate transgenerational rDNA copy number maintenance” describes single cell analysis and genetic analysis implicating Insulin signaling in controlling rDNA magnification in the *Drosophila* male germline. This is an interesting paper that demonstrates new molecular connections between a classical chromatin biology phenomenon and metabolic control.

We thank the reviewer for their thoughtful comments. Below we outline the revisions made in response to each of their comments.

The presentation of the data raises some questions which should be addressed in the text. The authors use RNAi to test candidates for effects on magnification, but mainly discuss only one - the Insulin Receptor. However in the supplementary table, four lines targeting *InR* were tested, and only one increased magnification. These others should be mentioned and explained. Is it possible that the one line that works has off-target effects?

Response: We agree with the reviewer about the importance of confirming the results with RNAi constructs by other methods. Please note that we tested 19 genes using 28 RNAi construct in our initial screening, making it difficult to confirm all of them with multiple methods. Thus, we used RNAi screening just as a primary screening method to concentrate candidates, followed by additional experiments to confirm that the results are not due to off-target effects. For this reason, we used expression of the dominant negative *InR^{K1409A}* allele to confirm the *InR* RNAi effects and for all further experiments.

We rather believe that the RNAi constructs without any effects are due to inefficient knockdown, which is often the case with *Drosophila* RNAi. Because RNAi is achieved by expressing short hairpin RNAi transgene in animal, each construct can target only ~25 nucleotides of target mRNA, which may or may not work and each construct has to be empirically tested for its efficacy. Thus, it is common that *Drosophila* RNAi lines do not work, requiring to try multiple constructs and validate by other methods, particularly when no effect is observed from the RNAi.

In case of *InR*, because two RNAi lines (out of 4 total) exhibited ectopic magnification, we considered it as a strong candidate, and thus confirmed the effects on rDNA magnification and R2 expression with expression of the well-documented dominant negative *InR^{K1409A}* construct (Wu et al., 2005) (**Fig 2B, H**). We also used constitutive active *InR^{K414P}* construct and observed the opposite effects (**Fig 2B, H**). Importantly,

after the initial RNAi-based candidate testing, we only used these mutant constructs for further *InR* experiments (**Fig 2C-H and Fig S4A**), avoiding potential concerns from off-targets.

We also would like to note that expression of the *InR* RNAi lines we found to strongly (TRiP.JF01482) and weakly (TRiP.JF01183) induce rDNA CN expansion have been previously demonstrated to have similar effects as *InR* mutants in other tissues (Im et al., 2018; Wang et al., 2023), further supporting the specificity of these effects.

Additionally, the table lists 3 ribosomal proteins as substantially increasing magnification. This seems worthy of comments in the main text, as well as the other 6 factors that have substantial effects.

Response: We thank this reviewer for noticing this interesting observation. Indeed, we were very intrigued by the collection of RNAi that target multiple ribosomal proteins and induced magnification. This observation, as this reviewer astutely pointed out, may suggest that disrupted ribosome function or biogenesis induces rDNA magnification. Exactly what molecular state is used as the cellular readout of ‘low rDNA CN’ to induce rDNA magnification is a long-standing question in the field, and is an active topic of study in the Nelson lab. We have added a short comment remarking on these candidates in the discussion (lines 372-385), but do not wish to speculate too much until we have a better understanding of how ribosome status could regulate magnification. We are hoping to be able to obtain insights into this question in future studies. We have also added a comment mentioning that the additional candidates provide intriguing possibilities for future studies to further reveal pathways that regulate R2 expression and rDNA CN expansion (lines 372-373). We decided not to speculate on the activity of individual candidates due to space limitations and we also wish to solidify those candidates by more thorough experimentation before giving them more serious consideration.

The figure legends and labeling should be expanded with more detail. The figures refer to “normal rDNA”, “Low rDNA”, “normal rDNA + *InR* RNAi”, “low rDNA + *InR* CA”. These should be given as genotypes, and explained in the legends. Figures 3A, B, E should be more clearly labeled. Figure 4E and F seem to be missing a sample each.

Response: We thank the reviewer for pointing out the need for more detailed genotypes. We revised the figure legends to include the full genotype and explanation for each control and experimental condition for clarity. However, we decided to keep simplified nomenclature within the figure to make them intuitive and easily digestible for the broad readership of *Nature Communications*. Regarding Figure 4E and 4F, the SY5 diet serves as a control for each rDNA condition. The low calorie SY1 diet is tested only in the normal rDNA condition (for its ability to induce rDNA magnification and GSC R2 expression), and the high calorie SY30 diet is tested only in the low rDNA condition (for suppression of rDNA magnification and GSC R2 expression). There are not any missing samples, as these are each separate independent experiments. We have revised the figure to better represent that these are separate experiments by adding a dotted line

between the normal and low rDNA conditions.

Reviewer #2 (Remarks to the Author):

In this compelling manuscript, Nelson et al extend upon their previous work identifying a critical role for the R2 transposon in regulating rDNA copy number to connect R2 regulation with upstream signaling pathways and physiological responses to nutrient sensation. The authors use a combination of RNA sequencing and genetic approaches to identify InR as a critical regulator of rDNA CN. Thorough analysis of potential downstream pathways identified mTOR1 as the proximate regulator of R2 expression upon InR inhibition. Finally, the authors find that dietary conditions, known to influence InR activity, ALSO controls R2 expression and regulation of rDNA CN. This is a high impact, exceptionally well-written manuscript with rigorous, clear results. Upon addressing some minor issues, I am enthusiastic about publication of this manuscript.

- The authors beautifully show the pathway from InR inhibition due to as yet unidentified mechanisms induced by low rDNA CN, through mTOR1 and effects on R2 expression. In addition, they clearly show effects of dietary conditions on regulation of R2 expression independent of the rDNA CN state of GSCs.

For this reviewer, what is missing are experiments related to the consequence of this dietary regulation over rDNA CN. There are several known consequences to GSCs and early spermatogonia due to changes in nutrition, including work from the Yamashita lab showing decreased GSCs due to starvation followed by de-differentiation to restore the stem cell pool following refeeding. It would be interesting to know whether forced depletion of R2 under starvation conditions alters the ability of the testis to appropriately restore homeostasis upon refeeding. Likewise, does inhibition of InR and/or expression of R2 under well-fed conditions cause a phenotypic consequence? Additional data parsing the degree of overlap between the “endogenous” vs “dietary” responses impacting rDNA CN and the consequences of altering the impact of dietary regulation on the “endogenous” rDNA CN pathway would add significantly to the impact of this work.

Response: We thank the reviewer for bringing up these very important and interesting points, which we address individually:

- i) Does R2 depletion influence dedifferentiation induced by starvation?
The Matunis lab (Sheng and Matunis, 2011), Bach lab (Herrera and Bach, 2018), and our lab (Yang and Yamashita, 2015) have described various aspects of the effects of starvation on germline homeostasis, which induces dedifferentiation upon refeeding. It should be noted that these studies used complete protein starvation (sugar only diet), whereas our study uses reduced protein diet. These differences may be important, as previous studies from our lab found that a similar ‘low protein’ diet (similar to what is used in this study) does not reduce GSC number, and thus no dedifferentiation occurs upon returning to a normal diet (Roth et al., 2012). Furthermore, since

induced R2 expression and rDNA CN expansions is limited to GSCs (Nelson et al., 2023a), and we show that InR continues to repress rDNA magnification activity in low rDNA CN SG, we do not expect that inhibiting R2 would impact the dedifferentiation of SG or GBs into GSCs during recovery from starvation.

Intriguingly, we found that low rDNA CN SG that did dedifferentiate into GSCs also did not express R2 (**Fig S4B-D**, see below). This finding indicates that InR-mediated R2 repression in SG is maintained through dedifferentiation, further suggesting that inhibiting R2 expression would not impact dedifferentiation during recovery from starvation. Other GSC-specific cellular features are not retained in dedifferentiated GSCs (Cheng et al., 2008; Herrera and Bach, 2018; Yadlapalli and Yamashita, 2013). These features notably include non-random sister chromatid segregation (Yadlapalli and Yamashita, 2013), which is also required for rDNA CN expansion (Watase et al., 2022). Thus, the mechanisms of rDNA CN expansion appear to be tightly restricted to GSCs, and may not even be able to be reacquired during dedifferentiation. The inability for these re-established GSCs to regulate R2 may explain why rDNA expansion becomes active in GSCs when nutrients are low / InR is inhibited, acting as a pre-emptive measure prior to this rDNA maintenance mechanism being lost once GSCs are lost and replaced by dedifferentiation. We have included these findings and discussion in our revised manuscript at lines 205-209.

ii) Does inhibition of InR and/or expression of R2 under well-fed conditions cause a phenotypic consequence?

In this study we demonstrate that the increase in rDNA CN is a phenotypic consequence of germline InR inhibition in well-fed (normal laboratory food) conditions (**Fig 2C**). Similarly, in our previous studies we found that transgenic R2 expression also induces ectopic rDNA CN increase, whereas a subset of cells lose rDNA CN (Nelson et al., 2023b), indicating unrestricted R2 expression causes global rDNA CN instability that can both increase and decrease rDNA, highlighting the importance for context-specific repression by insulin signaling.

iii) Parsing the degree of overlap between the “endogenous” vs “dietary” responses impacting rDNA CN and the consequences of altering the impact of dietary regulation on the “endogenous” rDNA CN pathway would add significantly to the impact of this work.

We thank the reviewer for raising this exceptionally important point. What the present study is showing is that diet has the ability to modulate the ‘set point’ of rDNA CN to trigger rDNA magnification. That is, the change in diet makes the cells perceive they need more or less rDNA than when they are on a normal diet, and they alter their rDNA CN expansion activity accordingly. What remains unclear is how diet influences the cell’s perception of its rDNA.

Based on the available data and knowledge, there is no reason to assume that there are two distinct 'endogenous' and 'dietary' pathways to modulate rDNA CN as the reviewer postulates. Instead, we propose that dietary changes alter the 'endogenous' pathway, because insulin signaling is an integral feature of this pathway. This alteration may occur either by shifting the physiological demand for rDNA (ie, the cell needs more or less rDNA than when it is on a normal diet) or alters the cells sensation of its rDNA (ie, the cell considers that it has more or less rDNA than it actually does and adjusts rDNA CN). We have concisely discussed this point in our manuscript (lines 321-325), but decided not to extensively describe potential mechanisms behind these possibilities since they remain purely speculative. An ability to separate these possibilities and fully understand how nutrients determine the rDNA CN 'set point' to trigger magnification relies on resolving the yet-to-be-identified mechanism of rDNA CN sensation. Fortunately, the findings of this study represent a major step towards achieving that end through revealing the role of IIS and mTor signaling in communicating the need for rDNA CN expansion. Furthermore, as discussed above in our response to reviewer #1, it is interesting to note that reduction in ribosomal proteins induce magnification, suggesting that cells' reduction of translation may be the trigger for the magnification. It is possible that altered translational activity resulting from dietary changes may underlie the diet-based impacts on rDNA magnification, indicating that it functions through the 'endogenous' rDNA sensation pathway. Further investigation of the role of ribosomal proteins in the regulations of rDNA CN expansion activity is an important next step to answering these questions, which we have emphasized in our manuscript (lines 398-408).

- This paper describes complicated interplays between signaling pathways and dietary conditions which is handled quite clearly in the text. One place where the outcomes of the work could be more clearly depicted is in the model shown in Fig.4. While the circular model showing connections between pathway components is compelling, including a more straight-forward and explicit set of diagrams to explain the impact of the work would be helpful. Specifically, showing the explicit outcomes of: 1. Normal food, effect of low rDNA; 2. Low protein food, high rDNA / low rDNA; 3. High protein food, high rDNA / low rDNA in terms of effects on InR/mTOR1 activity, R2 expression and changes (or not) in rDNA CN would be very helpful. And performing experiments suggested above would permit the authors to also include a phenotypic, tissue-level outcome for each of these conditions which would be quite compelling.

Response: We have updated the model in Fig 4G to more specifically model the pathway activities each genetic and dietary state discussed.

Reviewer #3 (Remarks to the Author):

In the manuscript entitled "Insulin signaling regulates R2 retrotransposon expression to orchestrate transgenerational rDNA copy number maintenance", Nelson and colleagues

explored the role of Insulin-mTORC1 axis in regulating ribosomal DNA (rDNA) copy number (CN) in *Drosophila* male germline stem cells. The authors utilized the single-cell RNA sequencing and single molecular RNA FISH to show that insulin signaling and mTOR repress the activity of rDNA-specific R2 retrotransposon, which in turn affect rDNA magnification. While the functions of R2 and Insulin/mTOR in rDNA magnification have been previously reported (PMC4401788 and PMC10266012), the current study's contribution to the new knowledge appears limited. Thus, it would not be recommended to be published on Nature Communications.

Response: We appreciate the reviewer connecting our previous findings on the function of R2 in rDNA magnification with findings from Aldrich and Maggert that increased IIS / mTor activity stimulates germline rDNA CN loss. Aldrich and Maggert, however, attributed the rDNA CN loss to increased instability, ie an increase in the occurrence of rDNA elimination events, and did not consider the possibility that IIS / mTor may regulate rDNA CN expansion activity or that disruption of rDNA CN recovery mechanism may explain their observed 'instability'. Moreover, they mainly focused on rDNA instability in somatic tissue. Although they also observed transgenerational inheritance of rDNA CN, Aldrich and Maggert did not address any of the mechanisms regarding how the rDNA CN is controlled in the **germline**, the tissue responsible for transgenerational rDNA CN inheritance (and changes). Whereas their work described that disrupted IIS and mTor activity can disrupt rDNA maintenance, our work reveals *why* this effect occurs. Importantly, our manuscript directly connects IIS / mTor to the regulator of rDNA CN in the **germline** (R2), and identifies that these pathways mediate the low rDNA CN 'cellular signal,' establishing their native role in rDNA CN maintenance, and not simply demonstrating the effect of their disruption. Therefore, our finding is not a re-discovery of what Aldrich and Maggert described: instead, the new findings in our manuscript prompt a reinterpretation of Aldrich and Maggert's data, revealing that the observed germline rDNA CN loss is the result of reduced rDNA CN expansion activity that is normally required to maintain the rDNA locus. Importantly, while the previous study suggested that the impact of IIS / mTor on rDNA CN is a secondary consequence of its impact on rRNA transcription, the current study reveals that this pathway is an integral, dynamic regulator of rDNA CN maintenance, providing the first insight into the cellular mechanisms that oversee this process. Therefore, this current study does not represent a simple extension of existing literature, but rather it provides a better, integrated model that explains Aldrich and Maggert's observations, and the observations of others. Together, our work establishes new fundamental features of rDNA maintenance, and describes the novel integration of a transposable element expression into the host's own gene regulatory pathways.

Major

1. The author used the bobbed score of offspring to quantify the rDNA magnification. There is a concern about potential overestimation of rDNA magnification due to a competitive advantage of normal rDNA CN sperms over low CN ones. It would indeed be beneficial if the authors could quantify the number of offspring produced by males with low vs. normal rDNA CN.

Response: We are grateful to the reviewer for pointing out the potential skewing of rDNA magnification frequencies due to possible differences in survival of progeny inheriting different amounts of rDNA. We addressed this concern by testing the viability of offspring from a subset of RNAi lines that succeed or failed in inducing rDNA CN expansion. Importantly, because the crosses to assess rDNA CN expansion use a female that is heterozygous for the complete rDNA deletion bb¹⁵⁸ chromosome (the X chromosome that does not have any functional rDNA copies, see Fig 2A), we can directly compare the viability of animals inheriting the bb¹⁵⁸ chromosome to their siblings inheriting a full rDNA locus from their mother. Through this analysis we found that there was a similar relative viability of bb¹⁵⁸-inheriting offspring among all tested RNAi lines, indicating that there is no competitive developmental or survival advantage in the offspring of the RNAi lines with observed rDNA CN expansion. This experiment and its description have been added to the revised manuscript in lines 160-163 and figure S3.

2. In Figure 2B and 2H, the penetrations of rDNA magnification and R2 expression in GSCs are not consistent, especially for the Normal rDNA + InR RNAi and Low rDNA groups (similarly in the Figure 3A,B, Rapamycin group). The author should clarify why a higher percentage of R2 expressing in low rDNA group leads to a lower percentage of rDNA magnification.

Response: This astute observation by the reviewer raises an intriguing question: why does the frequency of R2 expression not always correlate with the magnitude of rDNA magnification? There are two non-mutually exclusive considerations for this question. First, R2 expression is **unlikely** to be proportional to the degree of rDNA magnification. This is because, overly high expression of R2 is expected to 'reduce' (instead of 'increase') rDNA CN. R2 triggers rDNA magnification by generating DNA breaks at rDNA---if one break is made at the rDNA locus, it can induce magnification, but if more than one breaks are made on the rDNA locus, intervening copies of rDNA can be lost. Therefore, the lack of linear correlation between R2 expression level and the degree of rDNA magnification is not surprising. Second, whereas we have shown that R2 expression in GSCs is necessary and sufficient for rDNA magnification (Nelson et al., 2023b), it does not mean that R2 is *the only* factor that directs rDNA magnification. Our work demonstrates that R2 is downstream of the insulin pathway, but there are potentially many other layers of regulation also downstream of insulin (and perhaps upstream too). These potential regulators include decisions over: i) number of DSBs upon R2 expression, ii) DNA break repair pathway choice, iii) sister chromatid alignment during recombination, iv) sister chromatid segregation after USCE occurs. Therefore, the strong increase in rDNA magnification with a relatively weak increase in R2 expression observed by InR inhibition (**Fig 2B, H**), suggests that InR inhibition may stimulate both R2 expression and other programs that enhance the likelihood for a USCE event that increases rDNA CN. Conversely, the strong increase in R2 expression and relatively weak induction of rDNA magnification in rapamycin fed animals (**Fig 3A-B**) suggests that mTor represses R2 downstream of insulin signaling, but has little impact on these other effectors of USCE. Continued investigation into the differential gene expression between low and normal rDNA GSCs will provide an opportunity to

identify these potential mechanisms that may impact USCE outcomes during rDNA magnification.

3.To directly demonstrate that the InR pathway represses rDNA magnification through R2, the author should conduct a rescue experiment where R2 RNAi is used to suppress rDNA magnification in the normal rDNA + InR RNAi condition.

Response: Our revised manuscript now includes an experiment demonstrating that R2 RNAi indeed suppresses rDNA magnification in the normal rDNA + InR dominant negative condition (**Fig S4A**). We used the InR dominant negative instead of InR RNAi due to the concern for potential off target effects from the RNAi brought up by reviewer #1. Importantly, our control for this experiment co-expressed the dominant negative InR allele with GFP to control for potential suppression of rDNA magnification due to reduced expression of UAS transgenes from GAL4 being split between two UAS sites. The robust rDNA magnification in this control means that the suppression by expression of the R2 RNAi is due to R2 silencing.

4.Since nos-GAL4 is not exclusively expressed in germline stem cells (GSCs), using bam-GAL4 would be crucial for excluding the confounding effects from differentiated germ line cells to draw the conclusion that InR functions in GSCs.

Response: Our previous work demonstrated that R2 is not necessary in bam expressing germ cells for rDNA magnification (Nelson et al., 2023b), and that R2 expression and DSB forming activity is limited to GSCs in low rDNA conditions (Nelson et al., 2023a), indicating that the requirement to dynamically regulate R2 for rDNA maintenance is restricted to the GSCs. Intriguingly, we performed the suggested experiment and found that bam-Gal4 expression of the DN InR allele weakly induced rDNA CN expansion (**Fig S4B**). This finding reveals that InR also represses R2 expression in differentiating germ cells, and suggests that these cells are competent to induce some rDNA CN expansion if InR activity is repressed. However, our sequencing data indicates that InR expression is specifically reduced in GSCs when rDNA CN is low (**Fig 1D**). These data reveal that although all germ cells require InR to repress R2 expression, this repression is only relieved in GSCs when rDNA CN is low. This finding suggests that the mechanisms that regulate InR expression in response to rDNA CN are specific to GSCs. This specificity may be due to a difference in either the regulation of InR or an ability to 'sense' low rDNA CN (and subsequently suppress InR expression). Future investigations into these mechanisms will reveal the pathways the uniquely enable rDNA magnification in GSCs. This finding and discussion have been added to the manuscript in lines 192-205 and 394-396.

5.Providing comprehensive rDNA CN data beyond the bobbed score is essential for a full understanding of the rDNA magnification being studied. It would be better if the authors can provide all additional rDNA CN data besides the bobbed score as shown in Figure 2C.

Response: We have added figures 3B and S6B-C to include rDNA CN quantification in parallel with bobbed score to determine rDNA CN expansion activity.

Minor

- 1.All the *Drosophila* should be italicized.
- 2.Pi3K should be PI3K.

We thank the reviewer for pointing out these errors, which have been corrected in the revised manuscript.

Cheng J, Türkel N, Hemati N, Fuller MT, Hunt AJ, Yamashita YM. 2008. Centrosome misorientation reduces stem cell division during ageing. *Nature* **456**:599–604. doi:10.1038/nature07386

Herrera SC, Bach EA. 2018. JNK signaling triggers spermatogonial dedifferentiation during chronic stress to maintain the germline stem cell pool in the *Drosophila* testis. *eLife* **7**:e36095. doi:10.7554/elife.36095

Im SH, Patel AA, Cox DN, Galcko MJ. 2018. *Drosophila* Insulin receptor regulates the persistence of injury-induced nociceptive sensitization. *Dis Model Mech* **11**:dmm034231. doi:10.1242/dmm.034231

Nelson JO, Kumon T, Yamashita YM. 2023a. rDNA magnification is a unique feature of germline stem cells. *Proc Natl Acad Sci* **120**:e2314440120. doi:10.1073/pnas.2314440120

Nelson JO, Slicko A, Yamashita YM. 2023b. The retrotransposon R2 maintains *Drosophila* ribosomal DNA repeats. *Proc Natl Acad Sci* **120**:e2221613120. doi:10.1073/pnas.2221613120

Roth TM, Chiang C-YA, Inaba M, Yuan H, Salzman V, Roth CE, Yamashita YM. 2012. Centrosome misorientation mediates slowing of the cell cycle under limited nutrient conditions in *Drosophila* male germline stem cells. *Mol Biol Cell* **23**:1524–1532. doi:10.1091/mbc.e11-12-0999

Sheng XR, Matunis E. 2011. Live imaging of the *Drosophila* spermatogonial stem cell niche reveals novel mechanisms regulating germline stem cell output. *Development* **138**:3367–3376. doi:10.1242/dev.065797

Wang Y, Lopez-Bellido R, Huo X, Kavelaars A, Galcko MJ. 2023. The Insulin receptor regulates the persistence of mechanical nociceptive sensitization in flies and mice. *Biol Open* **12**:bio059864. doi:10.1242/bio.059864

Watase GJ, Nelson JO, Yamashita YM. 2022. Nonrandom sister chromatid segregation mediates rDNA copy number maintenance in *Drosophila*. *Sci Adv* **8**:eabo4443. doi:10.1126/sciadv.abo4443

Wu Q, Zhang Y, Xu J, Shen P. 2005. Regulation of hunger-driven behaviors by neural ribosomal S6 kinase in *Drosophila*. *Proc Natl Acad Sci* **102**:13289–13294. doi:10.1073/pnas.0501914102

Yadlapalli S, Yamashita YM. 2013. Chromosome-specific nonrandom sister chromatid segregation during stem-cell division. *Nature* **498**:251–254. doi:10.1038/nature12106

Yang H, Yamashita YM. 2015. The regulated elimination of transit-amplifying cells preserves tissue homeostasis during protein starvation in *Drosophila* testis. *Development* **142**:1756–1766. doi:10.1242/dev.122663